# Integration of Physiological, Transcriptomic and Metabolomic Reveals Molecular Mechanism of *Paraisaria dubia* Response to Zn^2+^ Stress

**DOI:** 10.3390/jof9070693

**Published:** 2023-06-21

**Authors:** Yue Wang, Ling-Ling Tong, Li Yuan, Meng-Zhen Liu, Yuan-Hang Du, Lin-Hui Yang, Bo Ren, Dong-Sheng Guo

**Affiliations:** School of Food Science and Pharmaceutical Engineering, Nanjing Normal University, Nanjing 210023, China

**Keywords:** *Paraisaria dubia*, Zn^2+^ stress response, transcriptomic, metabolomic, metal ion transport, microcycle conidiation

## Abstract

Utilizing mycoremediation is an important direction for managing heavy metal pollution. Zn^2+^ pollution has gradually become apparent, but there are few reports about its pollution remediation. Here, the Zn^2+^ remediation potential of *Paraisaria dubia*, an anamorph of the entomopathogenic fungus *Ophiocordyceps gracilis*, was explored. There was 60% Zn^2+^ removed by *Paraisaria dubia* mycelia from a Zn^2+^-contaminated medium. To reveal the Zn^2+^ tolerance mechanism of *Paraisaria dubia*, transcriptomic and metabolomic were executed. Results showed that Zn^2+^ caused a series of stress responses, such as energy metabolism inhibition, oxidative stress, antioxidant defense system disruption, autophagy obstruction, and DNA damage. Moreover, metabolomic analyses showed that the biosynthesis of some metabolites was affected against Zn^2+^ stress. In order to improve the tolerance to Zn^2+^ stress, the metabolic mechanism of metal ion transport, extracellular polysaccharides (EPS) synthesis, and microcycle conidiation were activated in *P. dubia*. Remarkably, the formation of microcycle conidiation may be triggered by reactive oxygen species (ROS) and mitogen-activated protein kinase (MAPK) signaling pathways. This study supplemented the gap of the Zn^2+^ resistance mechanism of *Paraisaria dubia* and provided a reference for the application of *Paraisaria dubia* in the bioremediation of heavy metals pollution.

## 1. Introduction

Human activities such as mining, smelting, urban waste treatment, and electroplating continuously release heavy metal pollutants into water and soil, damaging the ecological environment [1]. In recent years, the issue of heavy metal pollution has been constantly emerging, and the problem of Zn^2+^ pollution has gradually become apparent, arousing people’s attention. A high concentration of Zn^2+^ (16 mg/g) was found in soil near the smelting site [2]. Total soil Zn^2+^ concentrations in excess of 3 mg/g dry soil have been reported in contaminated agricultural fields [3,4]. The guideline values of Zn^2+^ in agriculture soil range from 120 to 250 mg/kg reported by various environmental departments (CCME, USEPA, EPA, etc.) [5]. Zn^2+^ is an essential trace element for plant growth and development, and the optimal Zn^2+^ level for healthy plants is 20–60 mg/kg dry weight [6]. However, above supra-optimal concentrations, this essential metal can have lethal effects on plant biology. Excessive Zn^2+^ inhibits seed germination, plant growth, and root development, disturbs photosynthesis, reduces the relative water content, and competes with other ions to disturb the physiological balance [7,8,9]. In addition, at the cellular level, excessive Zn^2+^ affects membrane integrity and permeability, causing oxidative damage and can even result in cell death [10]. Zn^2+^ concentration within the range of 25–150 mg/kg in plants is sufficient/tolerable, but when Zn^2+^ concentration reaches 100–400 mg/kg dry weight, the yield of plants will be reduced [5]. Zn^2+^ pollution damages plant growth, destroys the ecological environment, and hinders agricultural production. Therefore, the management of Zn^2+^ pollution is of great significance. Mycoremediation is a promising and effective method to control metal pollution, which is based on the bioaccumulation and biodegradation of macrofungi to eliminate toxic pollutants from the environment. Macrofungi can survive and accumulate large amounts of heavy metals from polluted environments, such as lead, copper, cadmium, chromium, and hydrargyrum [11]. In addition, previous studies have found a significant increase in the frequency of entomogenous fungi in the soil microfungal community of heavy metal-contaminated soil [12], suggesting that entomogenous fungi may be an excellent potential biomaterial for cleaning up Zn^2+^ pollution.

Cordyceps is a unique group of macrofungi that parasitizes insect larvae and pupae. More than 750 species have been reported, belonging to the order Ascomycota and classified into three families: Cordycipitaceae, Ophiocordycipitaceae, and Clavicipitaceae [13,14,15]. *Ophiocordyceps* (Ophiocordycipitaceae) is one of the most populous genera of entomopathogenic fungi, of which *Ophiocordyceps gracilis* (*O. gracilis*) is well-known traditional Chinese medicine [16]. *O. gracilis* is mainly distributed in China, Europe, and the Americas, a development that is generating more and more attention due to its excellent medicinal value [16,17]. The mycelia of *Paraisaria dubia* (*P. dubia*) is the asexual stage of *O. gracilis*, which is artificially cultivated as a substitute for natural *O. gracilis* [18]. It has been reported that the content of heavy metals in wild *Cordyceps* exceeds the standard, such as plumbum, hydrargyrum, and chromium [19]. This phenomenon indicates that heavy metal pollution has extended to the uninhabited wild environment, and the environmental pollution problem is becoming increasingly serious. Further, it also indicates that *Cordyceps* has a high capacity for heavy metal enrichment and is an ideal biological material to remediate heavy metal pollution [20]. However, there are no reports on the use of *Cordyceps* for bioremediation of heavy metal pollution, as previous studies have primarily focused on bioactive substances (polysaccharides, active peptides, cordycepin, etc.), pharmacologic effects, and artificial cultivation of *Cordyceps* [21,22,23,24]. Furthermore, the understanding of the molecular mechanism underlying heavy metal tolerance in *Cordyceps* remains limited, hampering the application of *Cordyceps* in heavy metal pollution control. Transcriptomics and metabolomics technologies offer efficient means to uncover information regarding adverse substance-induced molecular perturbations in cells and tissues. Moreover, the integration of transcriptomics and metabolomics has been employed to elucidate the molecular mechanisms in various macrofungi, including *Lentinula edodes*, *Ganoderma lucidum*, and *Agrocybe aegerita* [25]. In this study, *P. dubia* was used to investigate its tolerance capacity to Zn^2+^ stress, thereby expanding the application of mycoremediation in heavy metal pollution. Furthermore, by utilizing transcriptomics and metabolomics, the molecular tolerance mechanism underlying Zn^2+^ stress was explored, aiming to fill the knowledge gap concerning the molecular level of *Cordyceps*’ tolerance to Zn^2+^ stress.

In this study, a comprehensive analysis of physiological response, transcriptomics, and metabolomics was performed to gain insight into the tolerance of *P. dubia* to Zn^2+^ stress. The objects of this study were to (a) evaluate *P. dubia’s* physiological response under Zn^2+^ stress by measuring dry mycelia biomass, intracellular polysaccharides (IPS), extracellular polysaccharides (EPS), microcycle conidiation (MC) and Zn^2+^ removal efficiency; (b) identify pivotal differentially expressed genes (DEGs), differential metabolites and their involved essential pathways; (c) reveal the molecular mechanisms of tolerance of *P. dubia* response to Zn^2+^ stress. This study provided valuable tolerance molecular information of *P. dubia* under Zn^2+^ stress, instructing its potential application of heavy metals remediation.

## 2. Materials and Methods

### 2.1. Fungal Strain and Zn^2+^ Treatment

The strain of *P. dubia* (CGMCC No. 20731) used in this study is preserved in the China General Microbiological Culture Collection Center. *P. dubia* was cultured in different liquid mediums. The control group was cultured in ordinary medium (2% glucose, 1% peptone, 0.2% KH_2_PO_4_, 0.1% MgSO_4_·7H_2_O) [16], while the treatment group was cultured in Zn^2+^-contaminated medium (2% glucose, 1% peptone, 0.2% KH_2_PO_4_, 0.1% MgSO_4_·7H_2_O, 0.1% ZnSO_4_·7H_2_O). A 250 mL shake flask with 100 mL medium was cultured at 20 °C, 120 rpm. Three biological replicates were conducted per treatment condition. After being cultured for 10 days, mycelia pellets were separated from the fermentation broth containing spores by 200 mesh press cloth. Then, the fermentation broth was centrifuged at 8000 rpm for 10 min to take the precipitate to obtain spore powder. The mycelia and spores were washed three times with deionized water. Then place them in a freeze-drying machine (SCIENTZ-10N, NingBo Scientz Biotechnology, Zhejiang, China) to dry at −25 °C, 12 Pa. The weight of mycelia was recorded as biomass. In addition, mycelia and spores were ground into powder for subsequent experiments.

### 2.2. Microscopic Morphological Analysis

Micromorphological characteristics of *P. dubia* were observed under a conventional optical microscope (Olympus BX43, Tokyo, Japan), scanning electron microscopy (SEM) (SU8010 Tokyo, Japan) and transmission electron microscopy (TEM) (H-7650, Tokyo, Japan). Samples for SEM and TEM analysis were prepared according to the method described by Bo [26].

### 2.3. Preparation of IPS and EPS

The mycelia powder (0.5 g) of Zn^2+^ stress and the control were used for the extraction of IPS, and their fermentation broth was used for the extraction of EPS. The extraction method was conducted as described before [16], with minor modifications, as follows. Mycelia powder was extracted three times with ultrapure water (1:20, *w*/*v*) at 90 °C for 4 h. The mycelia extracting solution and fermentation broth were precipitated with a four-fold volume of 95% ethanol at 4 °C for 16 h and centrifuged at 8000× *g* for 10 min to obtain IPS and EPS. Then, the total polysaccharide content was determined using the phenol-sulfuric acid method.

### 2.4. Zn^2+^ Concentration Determination and Removal Rate

The content of Zn^2+^ in the mycelia, EPS, and conidia was determined by inductively coupled plasma-optical emission spectrometry/mass spectrometer (ICP-OES/MS, Thermo ICAP PRO) [27]. The concentration of Zn^2+^ in the samples was obtained by measuring the standard solution of Zn^2+^ and comparing which standard curve. The removal rate of Zn^2+^ was calculated using the following equation:Zn^2+^ removal rate (%) = (C_2_ + C_3_ + C_4_)/C_1_ × 100%(1)

C_1_ was the initial content in fermentation broth (22.8 mg/100 mL). C_2_, C_3_ and C_4_ were the content of Zn^2+^ in mycelia, conidia, and EPS obtained from 100 mL fermentation broth, respectively.

### 2.5. Nontargeted Metabolomic Profiling Analysis

Freeze-dried mycelia of Zn^2+^ stress and control (*n* = 5) were ground into a fine powder in liquid nitrogen, after which 60 mg powder was weighed in a centrifuge tube. Afterward, 500 μL pre-cooled methanol: H_2_O (1:1 = *v*:*v*) was added, followed by vortexing for 30 s at 4 °C. The homogenates were subsequently centrifuged at 14,000× *g* and 4 °C for 10 min. All the supernatant was then transferred into a fresh 1.5 mL centrifuge tube, frozen, and dried under a vacuum. The residue was dissolved in 300 µL of 2-chlorobenzalanine (4 ppm) methanol aqueous solution (1:1, 4 °C). Finally, the supernatant was purified by passing through a 0.22-µm membrane filter for LC-MS/MS analysis.

Samples were analyzed by Ultimate 3000 UHPLC System (Thermo Fisher Scientific, Waltham, MA, USA) coupled to Q Exactive high-resolution mass spectrometer (Thermo Fisher Scientific, Waltham, MA, USA). Chromatographic separations were accomplished using an ACQUITY UPLC ^®^ HSS T3 (150 × 2.1 mm, 1.8 µm) (Waters, Milford, MA, USA). The column was maintained at 40 °C. The flow rate and injection volume were set as 0.25 mL/min and 2 μL, respectively. For LC-ESI (+)-MS analysis, the mobile phases consisted of (C) 0.1% formic acid in acetonitrile (*v*/*v*) and (D) 0.1% formic acid in water (*v*/*v*). Separation was conducted under the following gradient: 0~1 min, 2% C; 1~9 min, 2%~50% C; 9~12 min, 50%~98% C; 12~13.5 min, 98% C; 13.5~14 min, 98%~2% C; 14~20 min, 2% C. For LC-ESI (−)-MS analysis, the analytes were carried out with (A) acetonitrile and (B) ammonium formate (5 mM). Separation was conducted under the following gradient: 0~1 min, 2% A; 1~9 min, 2%~50% A; 9~12 min, 50%~98% A; 12~13.5 min, 98% A; 13.5~14 min, 98%~2% A; 14~17 min, 2% A. Mass spectrometer settings for full-MS were as follows: the sheath gas pressure was set as 30 arb, the aux gas flow was set as 10 arb, the capillary temperature was set as 325 °C, the spray voltage: 3.50 kV (positive ion mode)/−2.50 kV (negative ion mode), the m/z range of MS1 was set as 100–1000, the MS1 resolving power was set as 70,000 FWHM, the number of data-dependent scans per cycle was set as 10, the MS/MS resolving power was set as 17,500 FWHM, the normalized collision energy was set as 30%, and the dynamic exclusion time set as automatic.

After the detection and analysis of four *Cordyceps* products by LC-MS/MS, the total ion chromatograms of all the samples were extracted. The raw data were processed using the XCMS package in R (V3.3.2) for feature detection, and retention time correction and alignment. The metabolites were identified by accuracy mass (<30 ppm) and MS/MS data which were matched with databases (Metlin, massbank, LipidMaps, and mzcloud). Metabolites are relatively quantified by peak area normalization. After normalization, only ion peaks with relative standard deviations (RSDs) less than 30% in QC were kept to ensure proper metabolite identification. OPLS−DA was performed using SIMCAP13.0 (Umetrics AB, Umea, Sweden) to visualize the differential metabolites among the samples. The selection of differentially abundant metabolites in the four sample classes was based on the combination of a statistically significant threshold of variable influence on projection (VIP) values and *p* values obtained using a two-tailed Student’s *t*-test and, based on the normalized peak areas of the metabolites with VIP ≥ 1 and *p* ≤ 0.05 were considered as differential metabolites. R^2^X, R^2^Y and Q^2^ were used to evaluate the quality of the OPLS−DA mode. Metabolite profiling and metabolomics data analyses were executed by BioNovoGene Co., Ltd. (Suzhou, China).

### 2.6. Transcriptomic Analysis

Total RNA was extracted from mycelia of Zn^2+^ stress and control (*n* = 3) by using TRIzol^®^ Reagent (Plant RNA Purification Reagent for plant tissue). The integrity and purity of the total RNA were determined using a 2100 Bioanalyzer and quantified using the ND-2000. Only high-quality RNA samples (OD260/280 = 1.8–2.2, OD260/230 ≥ 2.0, RIN ≥ 8.0, 28S:18S ≥ 1.0, >1 μg) were used to construct the sequencing library. RNA purification, reverse transcription, library construction and sequencing were performed at the Shanghai Majorbio Bio-pharm Biotechnology Co., Ltd. (Shanghai, China). RNA-seq transcriptomic libraries were prepared using the Illumina TruSeqTM RNA sample preparation Kit. The BLAST2GO (http://www.blast2go.com/b2ghome, accessed on 11 September 2022) program was used to obtain GO annotations of unique assembled transcripts for biological processes, molecular function, and cellular components. Metabolic pathway analysis was performed using the Kyoto Encyclopedia of Genes and Genomes (KEGG, http://www.genome.jp/kegg/, accessed on 11 September 2022). In order to identify differentially expressed genes (DEGs) between two groups, the expression level of each transcript was calculated according to the transcripts per million reads (TPM) method. RSEM (http://deweylab.biostat.wisc.edu/rsem/, accessed on 11 September 2022) was used to quantify the abundance of target genes. Differential expression analysis was performed using DEGs with |log2(foldchange)| ≥ 1 and *p* ≤ 0.05 as criteria for significantly differentially expressed genes [28]. In addition, the GO functional enrichment and KEGG pathway analyses were conducted (significantly enriched DEGs with Bonferroni-corrected *p* ≤ 0.05) using Goatools (https://github.com/tanghaibao/Goatools, accessed on 11 September 2022) and KOBAS (http://kobas.cbi.pku.edu.cn/home.do, accessed on 11 September 2022).

### 2.7. Statistical Analysis

Each experiment was performed with three replicates. The results are presented as means ± standard deviation (SD). The GraphPad prism 8.0 software was used to conduct one-way ANOVA tests for inter-group comparison, and *p* ≤ 0.05 was considered statistically significant.

## 3. Results

### 3.1. Morphological Alterations of Mycelia and Reproduction of P. dubia under Zn^2+^ Stress

Under Zn^2+^ stress, the surface of mycelia pellets showed protuberances compared with the control group (Figure 1). Interestingly, microcycle conidiation (MC) was induced under Zn^2+^ stress, which produced a large number of conidia. MC is the phenomenon of directly repeated sporulation of filamentous fungi following sexual or asexual spore germination with no growth of vegetative hyphae or very weak hyphal growth, which generally appears to serve as a fail-safe mechanism that allows fungi to efficiently transfer resources from cellular growth to conidia under adverse conditions [29,30]. As shown in Figure 1A, the conidia began to appear in large numbers on the 4th day and reached their maximum abundance (1.3 × 10^7^/mL) on the 12th day under Zn^2+^ stress. On the contrary, there were almost no conidia in the fermentation broth of the control group at the later stage. At the initial stage of the control group, the mycelia produced branched, after which the top of the hyphal branches produced conidia. The mature conidia disassociated from the top of the hyphae. As shown in Figure 1B, conidia produce new conidia by budding or on their germ tube. By utilizing TEM technology, yeast-like budding processes were captured. A conidium is about to be shed from the mother, and the cracks are clearly visible. Another mature conidium is producing a pro-conidium, and the channel between the pro-conidia and the mother conidium has not yet closed. Overall, the conidia formed by MC were rod-shaped (6.6–9.4 × 2.7–5.2 μm) or globular (2.6–3.7 μm), with smooth cell walls, large nuclei, and large lipid droplets. These conidia had a complete cellular structure and were capable of germination and reproduction.

Here, we documented the entire MC cycle of *P. dubia*, along with normal conidia production, through microscopic imaging. MC, as a subsidiary cycle of *O. gracilis’s* complete life, was described for the first time (Figure 1C). The MC process consisted of four phases. In phase I, conidia expand and germinate into a mycelium. In phases II and III, the mycelia grow further and branch out, after which the primary conidia are produced at the tips of mycelial branches. In phase IV, the conidia produce a large number of secondary conidia through budding if exposed to Zn^2+^ stress. Finally, these conidia produced by MC germinate under suitable conditions and enter the next MC cycle. Alternatively, they undergo normal conidiation when cultured on a solid medium.

### 3.2. Physiological Characteristic and Zn^2+^ Removal Capacity of P. dubia under Zn^2+^ Stress

After 12 days of cultivation under Zn^2+^ stress, dry mycelia biomass, intracellular polysaccharides (IPS), extracellular polysaccharides (EPS), and conidia yield as well as Zn^2+^ removal capacity was measured to evaluate Zn^2+^ tolerance of *P. dubia* (Table 1). The mycelial biomass (6 ± 1.4 g/L) of *P. dubia* decreased significantly under Zn^2+^ stress, which was consistent with previous reports that mushroom biomass decreased dramatically under high-level heavy metal [25]. The yield of IPS (135.5 ± 5.3 mg/g) was similar to the control group (119.3 ± 6.6 mg/g). However, the yield of EPS (233 ± 14.8 mg/L) was increased and 1.28 times that of the control group (181 ± 13.5 mg/L). Additionally, Zn^2+^ levels in mycelia and EPS were 119 and 139 times (11.9 and 13.2 mg/g) higher than those in control, respectively. Furthermore, the conidia also contained higher Zn^2+^, with a concentration of 6.2 mg/g. Based on the Zn^2+^ content of the above three substances, we calculated that the removal rate of Zn^2+^ was about 60%. These results indicate that *P. dubia* has great potential to effectively remove Zn^2+^ from high concentrations in the Zn^2+^ environment.

### 3.3. Transcriptomic Sequencing and DEGs Identification

In order to explore the molecular response mechanism of *P. dubia* to Zn^2+^ stress, the total RNA was extracted from Zn^2+^ stress and control samples for transcriptomic analysis. A total of 54,816,258 clean reads were obtained from the six samples by transcriptomic sequencing, and the average Q20 and Q30 value, as well as GC contents in clean data, were 97.96%, 94.03%, and 58.35%, respectively. All clean reads were assembled into 22,663 transcripts with an average length of 3912.65 bp and an N50 of 5914 bp. These transcripts were further assembled into 8075 genes with an average length of 2514.75 bp and an N50 of 4396 bp (Appendix A). In order to obtain comprehensive information on gene functions, six databases were used to annotate these genes. A total of 5924 (73.59%) genes were annotated. Among the annotated genes, NR (5720, 70.84%) and eggNOG (5357, 66.34%) had the largest match, followed by the GO, Pfam, Swissprot, and KEGG databases (Appendix A). Through the DEG screening (|log2(foldchange)| ≥ 1 and *p* ≤ 0.05), 1533 DEGs (702 upregulated and 895 downregulated) in the mycelia of *P. dubia* were identified to be responsive to Zn stress (Figure 2A).

### 3.4. Functional Annotation and Enrichment of DEGs

GO annotation analysis was performed to classify the DEGs functions. The results showed that the metabolic process and cellular process were the most enriched terms in the biological process ontology, which indicated that the cellular metabolism of *P. dubia* was affected under the condition of Zn^2+^ stress. Membrane part, cell part, and organelle were the most enriched terms in the cellular component ontology, which indicated that the membrane structure and mycelia growth of *P. dubia* were affected by Zn^2+^ stress. Catalytic activity and binding were the most enriched terms in the molecular function, which indicated that *P. dubia* may be resistant to Zn^2+^ stress by regulating, catalyzing, and activating various enzymes (Appendix A). The GO enrichment analysis was carried out to elucidate the specific biological functions of the DEGs. The upregulated DEGs were mainly enriched in the integral component membrane, intrinsic component membrane, hydrolase activity, acting on glycosyl bonds, and chitin-binding. Nevertheless, the downregulated DEGs were mainly enriched in the mitochondrion, nuclease activity, oxidation-reduction process, and generation of precursor metabolites and energy (Figure 2C,D). In order to confirm the biological pathways induced by Zn^2+^ stress, all DEGs were assigned to the KEGG database for KEGG pathway enrichment analysis. The upregulated DEGs were enriched in arginine and proline metabolism, tyrosine metabolism, tryptophan metabolism, and amino sugars and nucleotide sugars metabolism, whereas the downregulated DEGs were mainly enriched in oxidative phosphorylation, autophagy, steroid synthesis, longevity regulating pathway, and mitophagy (Figure 2E,F). Moreover, in order to reflect gene regulation more comprehensively, gene set enrichment analysis (GSEA) analysis was conducted. The results from it were similar to those in GO and KEGG enrichment analysis. The DEGs related to cellular respiration, respiratory electron transport chain, energy derivation by oxidation of organic compounds, cytochrome-c oxidase activity, and NADH dehydrogenase (quinone) activity were down-regulated (Figure 2B). Therefore, these pathways, glycolysis, tricarboxylic acid cycle (TCA), oxidative phosphorylation, antioxidant defense system, autophagy, metal ion transport, EPS synthesis, and amino acids metabolism, which should be connected by Zn^2+^ stress.

### 3.5. Critical Functional Genes in Response to Zn^2+^ Stress

According to the above enrichment analysis, we screened the critical genes related to oxidative phosphorylation, glycolysis, tricarboxylic acid cycle (TCA), antioxidant defense system, autophagy, metal ion transport, EPS synthesis, and amino acids metabolism (Appendix A). In detail, 31, 29, 12, and 12 DEGs were identified, which participate in the oxidative phosphorylation pathway, glycolysis and TCA cycle pathway, enzyme antioxidant system, and autophagy, respectively. Further, almost of those DEGs were downregulated under the condition of Zn^2+^ stress (Appendix A). Moreover, 14 genes associated with DNA repair and replication were identified, half of which were downregulated. (Appendix A). These results indicated that the cells were damaged in response to Zn^2+^ stress. A series of genes responsible for metal ion transport were expressed differently in mycelia between the Zn^2+^ stress and control groups. A total of 37 genes belonging to the ZRT/IRT-like protein transporter (ZIP), ATP-binding cassette transporter (ABC), heavy metal ATPases (HMAs), major facilitator superfamily general substrate transporter (MFS), and efflux pumping were significantly upregulated (Appendix A). A total of 51 genes were identified as participating in EPS synthesis, of which 34 genes were upregulated (Appendix A). It is noteworthy that 29 genes regulating MC formation (e.g., *brlA*, *abaA*, *ste12*, *vosA*, *PpoA*, *ppoC*, *atg3*, etc.) were induced to express under Zn^2+^ stress (Appendix A), which may be the important mechanism for *P. dubia* to tolerance Zn^2+^ stress.

### 3.6. Metabolic Changes of P. dubia under Zn^2+^ Stress

To elucidate the response of *P. dubia* under Zn^2+^ stress, we conducted a comparative analysis of the metabolites between the Zn^2+^-treated and control groups by untargeted metabolomics (LC−MS/MS), aiming to confirm differential metabolites associated with Zn^2+^ stress. The score plots of the OPLS−DA model (a multivariate analysis) showed considerable separation between the Zn^2+^ treated and control groups. The model parameters R^2^ and Q^2^ values were 0.91 and 0.06, respectively, indicating that the OPLS−DA model had an excellent predictive ability. The results indicated that the metabolites composition of *P. dubia* had significantly varied under Zn^2+^ stress (Figure 3A,B). A total of 612 metabolites were identified. Metabolites with VIP ≥ 1 and *p* ≤ 0.05 were selected as differential metabolites, and 207 were identified as differential metabolites (Appendix A). These differential metabolites were divided into seven categories, including amino acids and derivatives (21.26%), carbohydrates and derivatives (6.76%), nucleosides and derivatives (11.59%), fatty acids (4.35%), organic acids (3.86%), vitamins (1.93%), and others (39.61%). The category of others mainly includes alcohols, alkaloids, phenols, and some secondary metabolites, such as mannitol, ribitol, D-xylitol, choline, betaine, protocatechuic acid, etc. (Figure 3C). A large number of carbohydrates and amino acids were downregulated under Zn^2+^ stress. However, a few amino acids were shown to be upregulated under Zn^2+^ stress. Most of them have physiological functions of regulating osmotic pressure and antioxidants, such as arginine, taurine, tryptophan, and tyrosine (Figure 3C, Appendix A). In addition, more than 50% of differential metabolites were upregulated in the categories of nucleosides, vitamins, and others under Zn^2+^ stress, including adenosine, 2-deoxyadenosine, inosine, biotin, mannitol, D-xylitol, ribitol, D-arabitol, tyrosol, betaine, choline, acetylcholine, etc. (Appendix A). These upregulation metabolites may contribute to resistance to the Zn^2+^ stress of *P. dubia*.

Furthermore, the KEGG enrichment pathways analysis was conducted on all differential metabolites to confirm the pivotal metabolic pathways of *P. dubia’s* response to Zn^2+^ stress. As shown in Figure 3D, differential metabolites were significantly enrichment in the biosynthesis of secondary plant metabolites, ABC transporters, biosynthesis of alkaloids, cAMP signaling pathway, linoleic acid metabolism, biosynthesis of cofactors, TCA cycle, and amino acids metabolism (such as arginine and proline metabolism, protein digestion and absorption, biosynthesis of amino acids).

## 4. Discussion

### 4.1. Zn^2+^ Stress Affected in Energy Metabolism, Oxidative Stress, and Autophagy

When metal ions are transported into the cell and heavily accumulated, the basic metabolism of the cell will be hindered, including the critical basic metabolism pathways for obtaining energy in microorganisms, such as glycolysis, tricarboxylic acid (TCA) cycle, and oxidative phosphorylation [31,32]. Here, genes involved in glycolysis and the TCA cycle were mostly downregulated, which may lead to an interruption in energy supply, thus explaining the reduction in mycelia biomass under Zn^2+^ stress (Figure 4A). Nearly all genes that participated in oxidative phosphorylation were downregulated (Figure 4B), which may block electrons transmission in the electron transport chain, and these excess electrons will directly or indirectly react with molecular oxygen to generate reactive oxygen species (ROS), causing an acceleration of cell damage [33]. In addition, almost all genes related to antioxidant enzymes (*SOD*, *CAT*, *POD*, and *GSH-Px*) were significantly downregulated. Simultaneously, the amount of glutathione (GSH) and its precursor glutamic acid decreased (Figure 4D), which suggested that *P. dubia* was likely to suffer from oxidative stress. GSH has antioxidant effects and can easily combine with heavy metals such as zinc, lead, and arsenic to eliminate the toxicity of heavy metals [34]. Interestingly, two genes involved in GSH metabolism were upregulated (Appendix A), which may be a stress response to the depletion of GSH to scavenge ROS and Zn^2+^. This phenomenon is consistent with previous research that GSH was clearly downregulated when yeast was exposed to a high concentration of Cu^2+^ [35]. ROS disrupts cell membranes and organelles, thereby affecting the growth of mycelia. Previous studies have shown that ROS can cause severe damage to DNA, including disrupting DNA repair and replication systems, which hinder gene expression and lead to the death of organisms [36]. In this study, the expression levels of genes related to DNA processes were significantly altered in response to Zn^2+^ stress. Nearly half of the DEGs involved in DNA repair and replication were downregulated (Appendix A). The DNA replication licensing factor MCM, a component of the replication helicase necessary for the initiation and extension of DNA replication in eukaryotic cells, was significantly downregulated [37]. The results indicated that Zn^2+^ induced DNA damage by inhibiting the DNA repair and replication enzyme activity and disturbing the DNA-related protein activity. However, other upregulated DEGs were associated with DNA replication and repair, which played a crucial role in repairing DNA damage (Appendix A). For example, a gene associated with the DNA damage checkpoint was upregulated, and related research revealed that once the DNA damage checkpoint is activated, it hinders the progression of the cell cycle until the DNA repair system resolves the damage [38]. Autophagy is activated in cells to promptly remove damaged cell membranes, organelles, and abnormal proteins to maintain cellular homeostasis and recycle metabolic components [39]. The *Atg1/13* kinase complex was identified as a key positive regulator of autophagy, which interacts with *Atg* protein to regulate autophagy, such as *Atg17*. When *Atg17* is absent, the autophagy that is formed becomes smaller. At the same time, PKA inhibits autophagy formation by phosphorylating *Atg1* [40]. In this study, the expression of *Atg1*, *Atg13*, and *Atg 17* was decreased, while *PKA* was increased, indicating that autophagy was inhibited (Figure 4C). Autophagy is a mechanism for cellular self-protection, and inhibition of autophagy may disturb cellular homeostasis and lead to apoptosis. The upregulation of the apoptosis-inducing factor (Appendix A) and the downregulation of autophagy in this study are consistent.

### 4.2. Lead-Induced Detoxification and Defense Mechanisms

In this study, the Zn^2+^ stress brings a series of harm to the mycelia of *P. dubia*, which disrupts energy metabolism, blocks electron transfer, damages the antioxidant system and DNA, obstructs autophagy, and even induces apoptosis. To enhance the tolerance of *P. dubia* capacity to Zn^2+^, metal ion transport, EPS synthesis, and microcycle conidiation were activated. In addition, the biosynthesis of some metabolites was upregulated to resistant Zn^2+^ stress (Figure 4E–H). These coping strategies allow *P. dubia* successfully grow in the high Zn^2+^ environment.

#### 4.2.1. Metal Ion Transport Response to Zn^2+^ Stress

The basic strategy to resist metal ion stress is to reduce the accumulation of metal ions in cells, such as active efflux, which is a typical mechanism for biomining bacteria to resist high concentrations of copper [41]. In this study, the expression of genes associated with ZRT/IRT-like protein transporter (ZIP), ATP-binding cassette transporter (ABC), heavy metal ATPases (HMAs), major facilitator superfamily general substrate transporter (MFS), and efflux pumping was affected (Figure 4E, Appendix A). ZIP is known to mediate cellular uptake, intracellular trafficking, and detoxification of heavy metal cations such as zinc, iron, cadmium, copper, manganese, cobalt, and nickel [42]. In the present study, two ZIP transporters were downregulated, which may decrease the cellular uptake of Zn^2+^. However, *ZIP9* and a ZIP transporter were upregulated, and previous research found that under Zn^2+^ deficiency, the expression of *ZIP9* was upregulated in the shoot and root of *Thlaspi goesingense* to promote Zn^2+^ accumulation [43]. These indicate that the ion transport mechanism of ZIP in response to Zn^2+^ stress is complex and requires further investigation. ABC participates in various cellular processes, such as mitochondrial function, peroxisome biogenesis, and the export of Fe/S clusters, particularly in detoxifying heavy metals and removing toxic catabolic compounds [44]. Here, five ABC transporters were upregulated, with *ABCG* and *ABCB* showing upregulation. It has been reported that they can chelate heavy metals such as Cd^2+^ [25], which may contribute to enhancing tolerance to Zn^2+^. Additionally, HMAs, MFS transporters, and efflux pumping are also responsible for maintaining proper metal homeostasis in cells, and some genes related to them were upregulated in this research. HMAs consume ATP to pump metals such as copper, zinc, cadmium, and lead across the membrane and are responsible for metal transport and detoxification [45]. Studies have shown that HMAs are involved in the active efflux of Pb^2+^ or Pb^2+^ containing toxic molecules across the plasma membrane, followed by sequestration into inactive organelles to avoid Pb^2+^-induced toxicity [46]. MFS transporters can secrete a variety of toxic compounds, such as organic and inorganic ions, to enhance tolerance to heavy metals by enhanced efflux [27]. The efflux pumping serves as the foundation for resistance against heavy metal ions, such as P-type pumps, which play a critical role in mobilizing trace metals across the plasma membrane. The increased efflux of Zn^2+^ is one of the survival mechanisms of *P. dubia* mycelia under Zn^2+^ stress.

#### 4.2.2. Exopolysaccharides Synthesis Promotion Response to Zn^2+^ Stress

Secreting EPS to scavenge Zn^2+^ in the environment is another mechanism for *P. dubia* to respond to Zn^2+^ stress. Previous studies have shown that EPS is an important biosorbent for removing heavy metals, and some microorganisms chelate heavy metal ions by synthesizing EPS to grow under high-concentration metal ions stress [47]. Our research data showed that EPS yield was enhanced, simultaneously genes involved in EPS synthesis showed varying degrees of upregulation under Zn^2+^ stress, such as *HK*, *PGM*, *GALE*, and *PMM*, which are key enzymes associated with EPS synthesis and conversion steps (Figure 4F). Hexokinase (*HK*) catalyzed glucose to produce glucose 6-phosphate, glucose 6-phosphate was further converted to glucose 1-phosphate by phosphoglucomutase (*PGM*), then the glucose-1-phosphate generates UDP-Glc under the action of UTP-glucose-1-phosphate uridylyltransferase (*UGP2*) [48]. Furthermore, UDP-glucose 4-epimerase (*GALE*) catalyzed the conversion of UDP-Glc to UDP-Gal, the upregulated of *GALE* indicating that Zn^2+^ stress may have altered the monosaccharide composition of polysaccharides [49]. In addition, mannose 6-phosphate isomerase (*MPI*), phosphomannomutase (*PMM*), and mannose 1-phosphate guanylyltransferase (*GMPP*) gradually catalyzed Fructose-6P conversion to GDP-Man [50]. Hence, the upregulation of PMM may promote the generation of GDP-Man. Except for the upregulation of glycosyl donor synthetase, a large number of glycosyltransferases (GTs) and MFS were upregulated, which contribute to the polymerization of polysaccharides and secretion of polysaccharides out of the cell, respectively. Herein, the expression levels of these genes may determine the accumulation of polysaccharides under Zn^2+^ stress. This provides a reference for future research on the response mechanism of polysaccharide biosynthesis under heavy metal stress.

#### 4.2.3. Metabolic Response to Zn^2+^ Stress

KEGG analysis showed that differential metabolites were significantly enriched in plant secondary metabolite biosynthesis and alkaloid biosynthesis. Abiotic stress has been reported to induce secondary metabolites to form a mechanical barrier involved in osmotic stress resistance and ROS [45]. KEGG analysis showed that differential metabolites were significantly enriched in the biosynthesis of secondary plant metabolites and biosynthesis of alkaloids (Figure 3D and Figure 4G). Studies have shown that fungi protect themselves from abiotic stresses such as high temperature, drought, and salinization by accumulating compatible solutes, which are mostly sugars, sugar alcohols (polyols), betaine, and amino acids [51]. Several fungal spores are rich in mannitol, exhibiting excellent stress resistance. For example, the conidia of *Aspergillus niger* contain a large amount of mannitol, accounting for 10.9% of the dry weight [30]. In this study, we have found that mannitol, betaine, and five kinds of alcohol (D-Xylitol, ribitol, D-Arabitol, myo-Inositol, and tyrosol) were upregulated, which may contribute to against Zn^2+^ stress by regulating osmotic. In addition, pantothenic acid, biotin, niacinamide, nicotinic acid, andrographolide, and protocatechuic acid were upregulated, which may serve as compensatory mechanisms for the deficiency of antioxidant enzyme and GSH, given their excellent antioxidant capacity [45]. In addition, the level of aromatic amino acids (tryptophan and tyrosine), arginine, and taurine were increased. Furthermore, amino acid metabolism (arginine and proline metabolism, biosynthesis of amino acids, phenylalanine metabolism, and so on) was enriched. This result is consistent with the KEGG enrichment analysis in transcriptomics. Amino acids metabolism is closely related to secondary metabolism, which may provide precursors for the formation of secondary metabolites [52].

#### 4.2.4. Reproduction Response Based on MC in Zn^2+^ Stress

Apart from these aspects discussed above, the mycelia of *P. dubia* produced a large number of conidia through MC (Figure 4H). Conidia are more tolerant of adverse environmental conditions, and MC is a crucial survival tactic for *P. dubia* under Zn^2+^ stress. There are many factors that can induce the MC of fungi, such as nutrient deficiency, pH, light, and temperature [53,54,55]. While metal ions induce MC in fungi was found for the first time. In contrast to normal conidiation with a definite formation mechanism, the molecular mechanism of MC is still unclear, even though it was first observed in 1890 and has been reported in more than 100 fungal species. Only some research revealed the mechanism of MC preliminarily in some lower fungi which can’t form fruiting bodies, such as *Metarhizium anisopliae* and *Fusarium graminearum* [56,57]. Although the MC phenomenon has been reported in natural *C. militaris*, its mechanism has not been investigated [58]. Therefore, it is necessary to understand the formation mechanisms of MC. The underlying mechanism of MC of *P. dubia* was preliminarily analyzed, as shown in Figure 5.

**Signal-transduction of MC.** Conidiation is controlled by upstream regulatory genes which respond to the external environment. Previous studies have revealed some upstream regulatory genes, such as *fluG*, *flbA*, *flbB*, *flbC*, *flbD*, and *flbE*, which participate in regulating the core regulatory pathway of conidiation (*brlA* → *abaA* → *wetA*) in the model fungus *Aspergillus* [59]. In the current study, only *flbA* was identified and had no significant difference, suggesting that the upstream regulatory factor of MC of *P. dubia* is special, which is different from some filamentous fungi that have been reported. The mitogen-activated protein kinase (MAPK) signaling pathway plays important roles in fungal growth and development, such as cell cycle, morphogenesis, stress resistance, virulence, cell wall rigidity and integrity, intercellular signal transduction, etc. In addition, some studies have also shown that the MAPK signaling pathway regulates development and is required for conidiation [60]. Based on the MAPK signaling pathway (map04011) of yeast, we suggest a signal transduction model of MC, in which *abaA* was upregulated by downregulated *ste12* in the MAPK pathway to produce conidia, which consistent with other research conclusions that deletion of the *ste12* gene could promote conidia production [61]. Based on previous studies, ROS is a critical cell differentiation signal promoting conidiation and sexual development, which accumulation can promote conidiation in *N. crassa* [62]. In this study, Zn^2+^ damages the oxidation-reduction system and leads to the accumulation of ROS, which may be an inducer of MC. In addition, a total of 16 transcription factors (TF) were screened out, and a majority of TFs belong to the Zn cluster family (Zcf) (Appendix A). Zcf is a fungal-specific family of TF and is the largest family of TFs known in eukaryotes [63]. In taking these facts together, these data indicated that ROS may be an elicitor of MC, MAPK, and some factors which contain the zinc finger domain play an important role in regulating upstream regulation of MC.

**Genes involved in MC of *P. dubia* under Zn**^2+^ **stress.** The expression of the central regulators *brlA* and *abaA* were upregulated under Zn^2+^ stress, which contribute to MC formation. *BrlA* is a key TF whose overexpression in vegetative cells leads to growth cessation and the formation of viable conidia directly from the hyphal tips [59]. In addition, *brlA* contains a C_2_H_2_ zinc finger DNA-binding domain that recognizes *brlA* response elements in the promoter regions of certain genes, such as *abaA*. AbaA is a transcriptional enhancer factor 1 family TF governing the expression of *wetA*, *vosA*, and *velB* [64]. Previous research reported that *abaA* also affects secondary metabolism by regulating the expression of *veA*, *velB*, and *laeA* [65]. These were consistent with the metabolomic analysis that the biosynthesis of plant secondary metabolites and biosynthesis of alkaloids was significant enrichment. Therefore, this connection may link secondary metabolite biosynthesis with conidiation through a shared signal transduction pathway. The *wetA* gene activated by *abaA* completes the developmental role in the late stage of conidiation and plays an important role in conidia maturity [59], while *wetA* has no difference in this study. It has been reported that velvet regulator *vosA* couples with *velB* as a functional unit conferring conidia maturation and attenuating conidial germination in *A. nidulans* [66]. In this study, *vosA* significantly upregulated, which may represent a compensatory mechanism contributing to the conidia maturation. Except for the above core genes regulating MC, *PpoA*, *ppoC*, and *atg3* may also participate in regulating MC formation. *PpoA* and *ppoC* are dioxygenases that jointly regulate the balance between conidia and ascospore formation. When *ppoA* is deficient, the proportion of conidia significantly increases, while when *ppoC* is deficient, the proportion of ascospores significantly increases [67]. Here, the downregulation degree of *ppoA* was much higher than that of *ppoC*, which was conducive to the formation of conidia. Autophagy has been reported to regulate conidiation. For example, when deletion *atg3* (E2-like enzyme) and *atg7* (E1-like enzyme), the conidia and mycelia were reduced considerably [68,69]. *Atg3* was significantly downregulated in this study, which may inhibit conidia production. However, experimental results showed that plenty of conidia were produced, this may be due to the presence of multiple pathways activating MC formation, and when one of the pathways was repressed, the repressive effect was offset by an increase in the expression of another pathway. In addition, cell walls play important roles in providing mechanical support and participating in conidiation [70]. Relevant studies have shown chitinases I and II contribute to the development of conidia, and a GPI-deficient strain of *Fusarium verticillioides* was not competent in conidiation and cell wall formation [71,72]. In this study, the expression level of chitinase I, GPI-anchored endo-1, and an uncharacterized GPI-anchored protein was significantly upregulated, which may facilitate the formation of MC.

## 5. Conclusions

Based on this study, we are convinced that *P. dubia* is a potential strain for bioremediation of Zn^2+^ pollution. There were 207 differential metabolites, and 1533 DEGs were identified from *P. dubia* under the condition of Zn^2+^ stress by transcriptomic and metabolomic analysis, which mainly enriched in TCA cycle, oxidative phosphorylation, autography, cell membrane and organelle, secondary metabolites biosynthesis, and amino acids metabolism. From the overall analysis, the mycelia of *P. dubia* enhances its tolerance to Zn^2+^ stress through the following strategies, such as restricting Zn^2+^ entering the cell by downregulating ion transport, reducing bioavailable Zn^2+^ concentration through EPS chelation, regulating metabolites to ensure cell viability, cell osmotic balance, and ROS scavenging, and inducing the production of a large number of conidia to maintain population continuity. Moreover, the phenomenon of MC was first observed in the liquid fermentation of *Cordyceps*. The MC may be an intelligent strategy employed by entomopathogenic fungi to resist adverse environments, which appear to be activated by increasing ROS levels. This study will provide a new strategy for the bioremediation of Zn^2+^ pollution and serve as a reference for the investigation of the heavy metal tolerance mechanism in *Cordyceps*.

## Figures and Tables

**Figure 1 jof-09-00693-f001:**
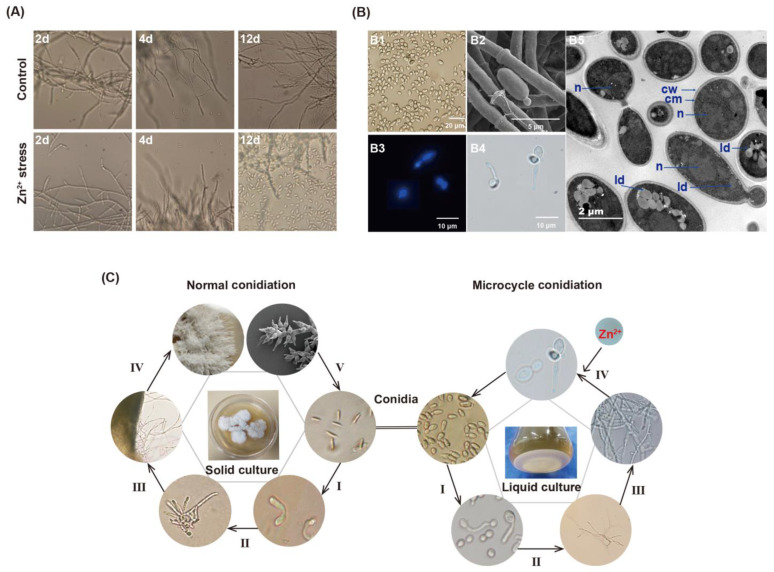
MC condition of *P. dubia* induced by Zn^2+^ stress. (**A**) The process of conidiation in control and Zn^2+^ stress. (**B**) Micrograph of the conidia produced by MC. (B1), (B2) brightfield micrograph; (B3) fluorescence micrograph; (B4) scanning electron micrograph; (B5) transmission electron micrograph showing the ultrastructure of microconidia of *P. dubia*. Abbreviations: n, nucleus; ld, lipid droplet; cw, cell wall; cm, cell membrane. (**C**) Asexual reproduction cycle of *P. dubia*.

**Figure 2 jof-09-00693-f002:**
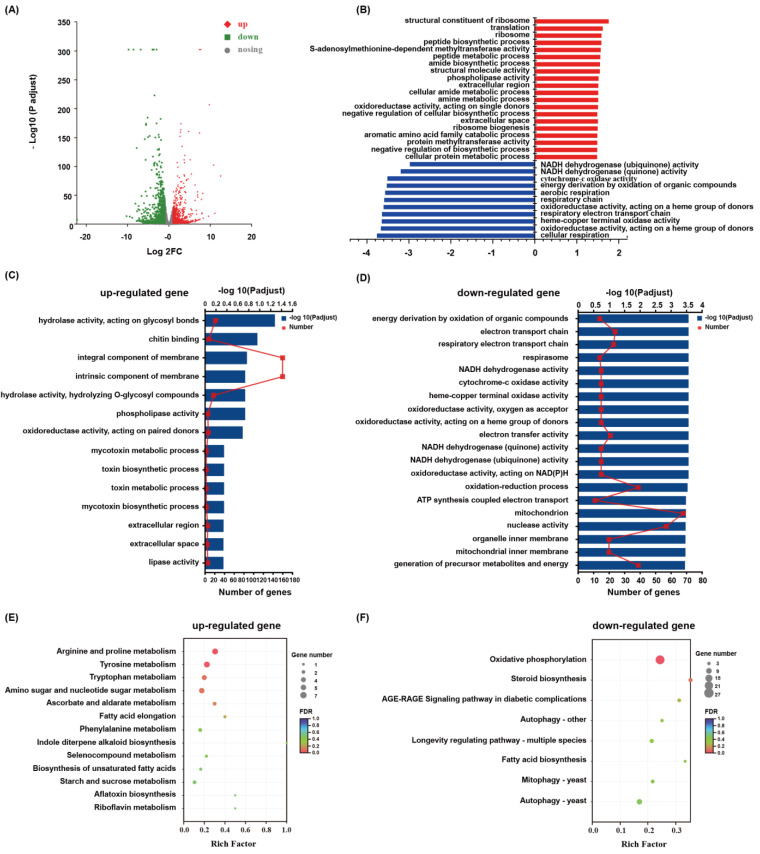
Distribution and function of DEGs related to Zn^2+^ stress. (**A**): Venn diagram and Volcano plot of DEGs; (**B**): GSEA analysis of Zn^2^ treatment; (**C**,**D**): Scatter plot of GO enrichment of up- and downregulated genes. (**E**,**F**): Scatter plot of enriched KEGG pathways of upregulated and downregulated genes. The Rich factor is the ratio of the number of genes that were differentially expressed to the total number of genes in a certain pathway. The figure shows the top 20 enriched pathways.

**Figure 3 jof-09-00693-f003:**
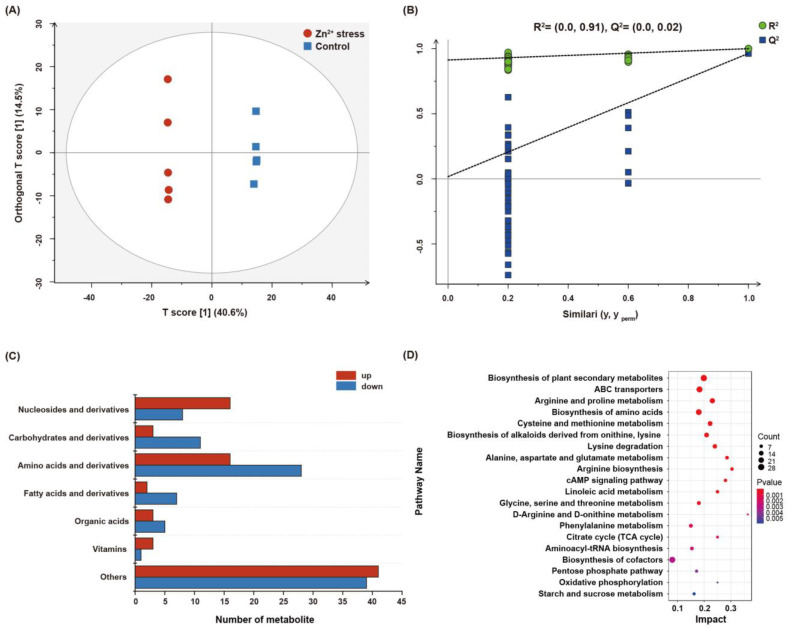
Metabolomic analysis of Zn^2+^ stress and normal cultivation of *P. dubia*. (**A**) OPLS−DA score plot. (**B**) Statistical validation of the OPLS−DA model with permutation. (**C**) The numbers of seven categories of differential metabolites. (**D**) KEGG pathway analysis of differential metabolites.

**Figure 4 jof-09-00693-f004:**
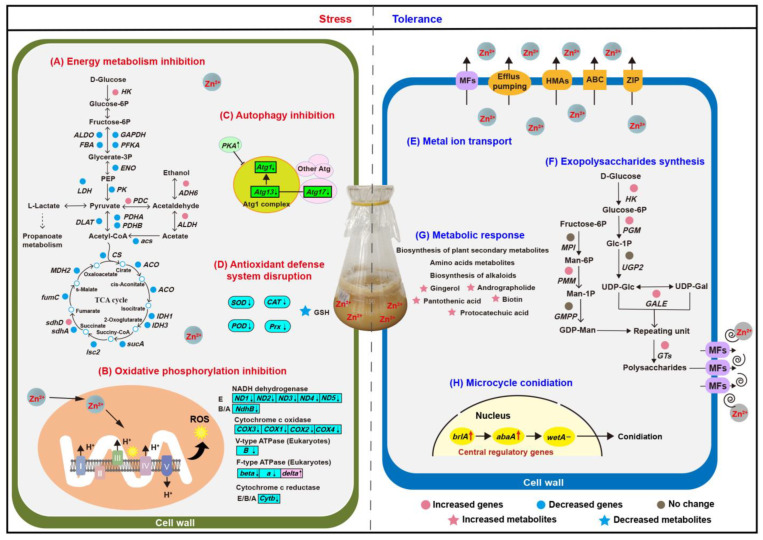
Zn^2+^ stress and tolerance mechanism in mycelia of *P. dubia*. (**A**) Energy metabolism inhibition. (**B**) Oxidative phosphorylation inhibition. (**C**) Autophagy inhibition. (**D**) Antioxidant defense system disruption. (**E**) Metal ion transport. (**F**) Exopolysaccharides synthesis. (**G**) Metabolic response. (**H**) Microcycle conidiation. ROS—reactive oxygen species; SOD—superoxide dismutase; CAT—catalase; Prx—peroxidase.

**Figure 5 jof-09-00693-f005:**
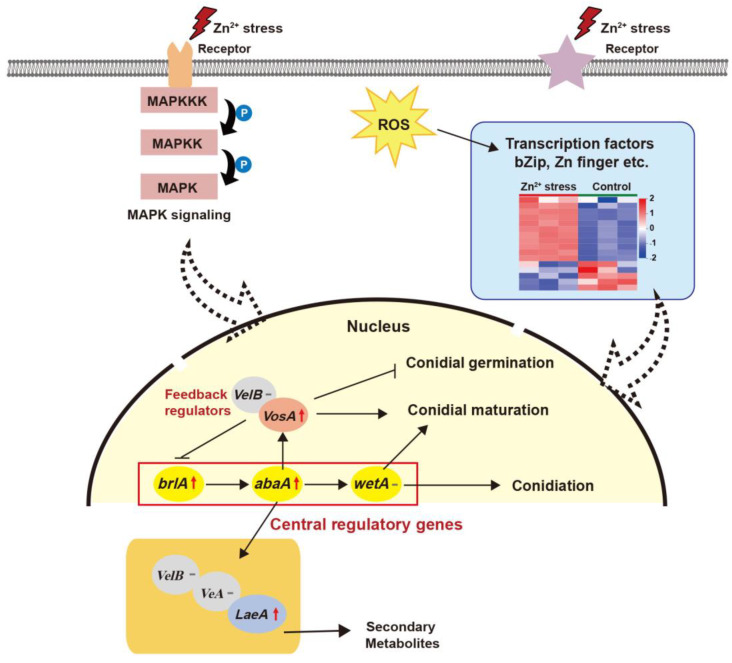
The predicted microcycle conidiation pathways induced by Zn^2+^ stress in *P. dubia*.

**Table 1 jof-09-00693-t001:** The response and Zn^2+^ accumulate ability of *P. dubia* under Zn^2+^ stress.

	Mycelia Biomass (g/L)	IPS (mg/g)	EPS (mg/L)	Conidia (g/L)	Zn^2+^ Concentration (mg/g)		
					Mycelia	EPS	Conidia
Zn^2+^ stress	6 ± 1.4 ^a^	135 ± 5.3 ^a^	233 ± 14.8 ^a^	0.162 ± 0.0062	11.9 ± 1.2 ^a^	13.2 ± 1.6 ^a^	6.2 ± 0.9
Control	13 ± 1.8 ^b^	119 ± 6.6 ^a^	181 ± 13.5 ^b^	ND	0.1 ± 0.004 ^b^	0.095 ± 0.003 ^b^	ND

“ND” means conidia not be detected. Means with different letters are significantly different at *p* ≤ 0.05.

## Data Availability

The RNA-seq data reported in the current study are available in the NCBI SRA database. BioProject number is PRJNA763311.

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
