# Peer review of "Integration of Physiological, Transcriptomic and Metabolomic Reveals Molecular Mechanism of Paraisaria dubia Response to Zn2+ Stress"

_jof, 2023, doi:10.3390/jof9070693_

Round 1
Reviewer 1 Report (Previous Reviewer 2)
The manuscript (jof-2443678) of Yue Wang, Ling-Ling Tong, Li Yuan, Meng-Zhen Liu, Yuan-Hang Du, Lin-Hui Yang, Bo Ren, Dong-Sheng Guo titled “Integrated physiological, transcriptomic and metabolomic analysis the response of Paraisaria dubia to Zn2+ stress” before publication in Journal of Fungi needs some minor corrections.
Comments:
Page 1
Introduction
Line 31 I suggest writing 16 mg/g instead of “16000 mg/kg”
Line 32 I suggest writing 3 mg/g instead of “3000 mg/kg”
Page 3
Line 97 should be “10 min” instead of “10min”
Line 145 should be “5 mM” instead of “5mM”
Page 5
Line 204 should be “1.3x107/mL” instead of “1.3x107/ml”
Page 11
Line 372 should be “Cu2+ “ or “Cu(II)” instead of “Cu (II)”
Page 16
Line 565 In the task: "Based on this study, we believe that P. dubia is a potential strain for bioremediation of Zn2+ pollution"
I suggest you enter: “we are convinced’ instead of “we believe”
Author Response
Dear Editor and Reviewers,
We sincerely appreciate for your constructive and helpful comments and suggestions to our manuscript entitled. We are pleased to answer the questions of the reviewers’ and the manuscript (Manuscript number: jof-2443678) has also been extensively revised according to the comments (resubmitted online).
The answers also have been summarized one by one as listed below.
Finally, we sincerely hope that the revised manuscript can meet the standard for publication in your journal.
PS: The modifications were marked in red in the revised manuscript.
Thank you very much.
Sincerely yours,
Dong-Sheng Guo
Reviewer 2 Report (New Reviewer)
The manuscript written by Wang et al. titled “Integrated physiological, transcriptomic and metabolomic analysis the response of Paraisaria dubia to Zn2+ stress” is novel and scientific. The presentation is easy to follow and informative about the use of microorganisms for heavy metal pollution control. Nevertheless, some corrections are required.
1) The English language of the MS needs minor checks.
2) The aim, statement problem and justification of the study need to be clearly stated in the introduction section.
3) The title can be modified to rhyme with the information provided in L85-86.
Minor comments.
L19 and 20: EPS, ROS, MAPK, write in full.
L24: Re-write the keywords to be more informative
L110: Rephrase the sentence
L128: Which sample?
Table 1: No statistical evidence shown.
The English language of the MS needs minor checks
Author Response
Dear Editor and Reviewers,
We sincerely appreciate for your constructive and helpful comments and suggestions to our manuscript entitled. We are pleased to answer the questions of the reviewers’ and the manuscript (Manuscript number: jof-2443678) has also been extensively revised according to the comments (resubmitted online).
The answers also have been summarized one by one as listed below.
Finally, we sincerely hope that the revised manuscript can meet the standard for publication in your journal.
PS: The modifications were marked in red in the revised manuscript.
Thank you very much.
Sincerely yours,
Dong-Sheng Guo
This manuscript is a resubmission of an earlier submission. The following is a list of the peer review reports and author responses from that submission.
Round 1
Reviewer 1 Report
Page 2, line 48: it is correct ambient environment sites?
Page 2, line 48: Nordgren et al. …? year?
INTRODUCTION: the argument that the authors use to work with zinc as a dangerous metal is not very convincing, since it is used a lot in everyday life.
Page 2, line 87: cine-enriched médium- is that correct?
Page 3, line 98: It is not correct to mention that the methodology was carried out as in his previous study, the methodology must be mentioned in any way and cited.
Page 3, line 102: ICP-OES/MS is not an atomic absorption spectrometer
Page 3, line 112: metabonomic, is that correct?
Page 4 and 5, line 181-210: the authors do not mention figures 1A and 1B in the paper
Almost all the figures have a very small size to be able to analyze them correctly
All the writing of point 3.3 does not have the same format as the other points.
The authors should try to make points 3.3 and 3.4 more friendly.
Page 13, line 525: metabolism et al., is that correct?
Reviewer 2 Report
The manuscript (jof-2279813) of Yue Wang, Ling-Ling Tong, Li Yuan, Meng-Zhen Liu, Yuan-Hang Du, Lin-Hui Yang, Bo Ren, Dong-Sheng Guo titled “Integrated physiological, transcriptomic and metabolomic analysis the response of Paraisaria dubia to Zn2+ toxicity”
Comments:
Page 1
Introduction
Line 44 should be zinc or Zn2+ instead of “Zn”
Page 2
Lines 47 and 48,
suggest the names of the elements or their cations
Line 87 should be “20°C” instead of “20 °C”
Page 3
The section 2.4. Zn2+ concentration determination and removal rate
Line 104 i 105
The description does not specify the type of apparatus used for lyophilization of mycelium and the temperature and vacuum values at which the lyophilization was performed and not given.
Line 115 should be “4°C” instead of “4 °C”
Line 116 should be “4°C” instead of “4 °C”
Line 124 should be “40°C” instead of “40 °C”
Line 134 should be “325°C” instead of “325 °C”
Line 145 should be “30%” instead of “30 %”
The section 2.5.
Line 112 should be “metabolomic” instead of “metabonomic”
The manuscript does not provide reference material used for the quantification of inorganic and organic chemical compounds.
Author Response
Detailed Response to Reviewers
Manuscript Number: jof-2279813
Manuscript Title: Integrated physiological, transcriptomic and metabolomic analysis the response of Paraisaria dubia to Zn2+ toxicity
Type of manuscript: Article
Dear Editor and Reviewers,
We sincerely appreciate for your constructive and helpful comments and suggestions to our manuscript entitled. We are pleased to answer the questions of the reviewers’ and the manuscript (Manuscript number: jof-2279813) has also been extensively revised according to the comments (resubmitted online).
The answers also have been summarized one by one as listed below.
Finally, we sincerely hope that the revised manuscript can meet the standard for publication in your journal.
PS: The modifications were marked in red in the revised manuscript.
Thank you very much.
Sincerely yours,
Dong-Sheng Guo
Response to Reviewer 2 Comments
Point 1: Page 1, Line 44 should be zinc or Zn2+ instead of “Zn”.
Response: We have corrected it in the revised manuscript (line 42).
Point 2: Page 2, Lines 47 and 48, suggest the names of the elements or their cations.
Response: Thank you for your reminder. The question has been amended in the revised manuscript (line 49).
Point 3: Page 2, Line 87 should be “20°C” instead of “20 °C”
Response: We have corrected it in the revised manuscript (line 94).
Point 4: Page 3, The section 2.4. Zn2+ concentration determination and removal rate. Line 104 i 105. The description does not specify the type of apparatus used for lyophilization of mycelium and the temperature and vacuum values at which the lyophilization was performed and not given.
Response: Thank you for your reminder. This information has been supplemented in the manuscript (lines 99-100).
Point 5: Line 115 should be “4°C” instead of “4 °C”; Line 116 should be “4°C” instead of “4 °C”; Line 124 should be “40°C” instead of “40 °C”; Line 134 should be “325°C” instead of “325 °C”; Line 145 should be “30%” instead of “30 %”.
Response: We have corrected it in the revised manuscript (lines 130, 131, 139, 149, 160).
Point 6: The section 2.5. Line 112 should be “metabolomic” instead of “metabonomic”
Response: Thank you for your reminder. The mistake has been corrected (line 127).
Point 7: The manuscript does not provide reference material used for the quantification of inorganic and organic chemical compounds.
Response: Thank you for your reminder. The method of Zn2+ concentration determination has been added in the revised manuscript (lines 120-121).
In order to comprehensively understand the effects of Zn2+ stress on P. dubia metabolism, untargeted metabolomic analysis was conducted in this study, Therefore, metabolites are relatively quantified by peak area normalization (line 158-159). We also added the relevant descriptions in the revised manuscript.
Reviewer 3 Report
The manuscript is well written and presented. This finding looks very useful for readers who are interested in use of fungi for removal Zn (II) from polluted environments.
Here are several comments that will improve the manuscript.
Line 11: Delete ‘(P. dubia)’.
Lines 21, 22: Change to ‘Paraisaria dubia’.
Lines 56, 57: Briefly discuss the characters, taxonomy and the importance of Paraisaria species.
Line 84: Delete ‘the anamorph of O. gracilis’.
Lines 85, 86: If there is a reference, add it.
Line 90: Correct ‘The micromorphology of P. dubia conidia and mycelium’ to ‘Micromorphological characteristics of P. dubia’.
Line 212 (Figure 1B): Correct “D1->D5’ to ‘B1->B5’.
Line 231 (Table 1): How did the authors measure biomass? ned to describe it.
